# The Mean Values of Character Sums and Their Applications

**Jiafan Zhang** and **Yuanyuan Meng** *

School of Mathematics, Northwest University, Xi'an 710127, China; zhangjiafan@stumail.nwu.edu.cn
* Correspondence: yymeng@stumail.nwu.edu.cn

**Abstract:** In this paper, we use the elementary methods and properties of classical Gauss sums to study the calculation problems of some mean values of character sums of special polynomials, and obtained several interesting calculation formulae for them. As an application, we give a criterion for determining that 2 is the cubic residue for any odd prime $p$.

**Keywords:** character sums of polynomials; mean value; classical Gauss sums; analytic methods; calculation formula





## 1. Introduction

Let $q > 1$ be an integer. For any positive integer $k > 1$, an integer $a$ with $(a, q) = 1$ is called $k$-th residue modulo $q$, if the congruence equation $x^k \equiv a(\bmod q)$ has at least an integer solution $x_0 \bmod q$. That is, $x_0^k \equiv a(\bmod q)$. If $k = 2$ and $q = p$, an odd prime, Legendre first introduced the quadratic character function, i.e., Legendre's symbol modulo $p$, which is defined as follows:

$$\left(\frac{a}{p}\right) = \begin{cases} 1, & \text{if } a \text{ is a quadratic residue modulo } p; \\ -1, & \text{if } a \text{ is not a quadratic residue modulo } p; \\ 0, & \text{if } p \mid a. \end{cases}$$

This function occupies a very important position in elementary number theory and analytic number theory. Many number theory problems are closely related to it. Some works can be found in references [1–13], which will not be listed here. For example, if $p$ is a prime with $p = 4k + 1$, then $p$ can be expressed as the sum of two squares of positive integers. That is, $p = \alpha^2(p) + \beta^2(p)$. More precisely (see Theorem 4–11 of [14]), we have the identity

$$p = \alpha^2(p) + \beta^2(p) \equiv \left(\sum_{a=1}^{\frac{p-1}{2}} \left(\frac{a^3 + ra}{p}\right)\right)^2 + \left(\sum_{a=1}^{\frac{p-1}{2}} \left(\frac{a^3 + na}{p}\right)\right)^2,$$

where $\left(\frac{*}{p}\right)$ denotes the Legendre's symbol modulo $p$, $r$ and $n$ are any two integers with $(rn, p) = 1$, such that $\left(\frac{rn}{p}\right) = -1$ or $\left(\frac{r}{p}\right) + \left(\frac{n}{p}\right) = 0$.

In addition, Legendre's symbol has many unique properties. For example,

$$\left(\frac{-1}{p}\right) = (-1)^{\frac{p-1}{2}}, \quad \left(\frac{2}{p}\right) = (-1)^{\frac{p^2-1}{8}} \text{ and } \left(\frac{q}{p}\right) \cdot \left(\frac{p}{q}\right) = (-1)^{\frac{(p-1)(q-1)}{4}},$$

where $p$ and $q$ are two different odd primes (see [15]).

In this paper, we concentrate our attention on the mean values of character sums of the polynomial $f(x, y)$. That is,

$$\sum_{b=0}^{p-1}\left|\sum_{a=0}^{p-1}\chi(f(a,b))\right|^{4},\tag{1}$$

where $f(x,y)$ is an integer coefficient polynomial of $x$ and $y$, $p$ is an odd prime, $\chi$ is any non-principal character modulo $p$.

Our aim is to give an exact formula for calculating the sum in (1). Of course, for a general polynomial $f(x,y)$, this is hard to do. However, for some special polynomials $f(x,y)$, we can still produce fairly good results. The main purpose of this paper is to illustrate this point. That is, we will use elementary methods and the properties of the classical Gauss sums to prove the following results.

**Theorem 1.** *Let $p$ be an odd prime with $p \equiv 1 \pmod 6$. Then, for any non-principal character $\chi$ mod $p$, we have the identity*

$$\frac{1}{p-1}\cdot\sum_{n=1}^{p-1}\left|\sum_{a=0}^{p-1}\chi\left(a^{3}+n\right)\right|^{4}=\begin{cases} 6p^{2}, & \text{if } \chi^{3}\neq\chi_{0};\\ p^{2}+4p+1, & \text{if } \chi^{3}=\chi_{0}, \end{cases}$$

*where $\chi_0$ denotes the principal character modulo $p$.*

**Theorem 2.** *Let $p$ be an odd prime with $p \equiv 1 \pmod 6$. Then, for any three-order character $\lambda$ mod $p$, we have the identity*

$$\frac{1}{p-1}\cdot\sum_{n=1}^{p-1}\left|\sum_{a=0}^{p-1}\lambda\left(a^{3}+n\right)\right|^{6}=(p+1)\left(p^{2}+8p+1\right)-d(d^{2}-3p),$$

*where $4p = d^2 + 27b^2$, $d$ is uniquely determined by $d \equiv 1 \pmod 3$ (see [3]).*

**Theorem 3.** *Let $p$ be an odd prime with $p \equiv 1 \pmod 6$. Then, we have*

$$\sum_{a=0}^{p-1}\left(\frac{a^{3}+2}{p}\right)=\left(\frac{2}{p}\right)\cdot d=(-1)^{\frac{p^{2}-1}{8}}\cdot d,$$

*where $\left(\frac{*}{p}\right)$ denotes the Legendre's symbol modulo $p$, $d$ is the same as the definition in Theorem 2.*

From Theorem 3, we may immediately deduce the following two corollaries.

**Corollary 1.** *Let $p$ be an odd prime with $p \equiv 1 \pmod 6$. Then, 2 is a cubic residue modulo $p$ if and only if $d$ is an even number.*

**Corollary 2.** *Let $p$ be an odd prime with $p \equiv 1 \pmod 6$. Then, 2 is a cubic residue modulo $p$ if and only if there are two integers $d_1$ and $b_1$ such that*

$$p=d_{1}^{2}+27\cdot b_{1}^{2},$$

*where $d_1$ is uniquely determined by $d_1 \equiv -1 \pmod 3$ (see G. Frei [11]).*

**Some notes.** Our results reveal the value distribution properties of the character sums of polynomials. Although the value distribution of individual sums is very irregular, its mean value shows good distribution properties, such as Theorem 1. In addition, our results seem to be generalized, which means that we can study the mean value

$$\sum_{n=1}^{p-1}\left|\sum_{a=0}^{p-1}\chi\left(a^{h}+n\right)\right|^{4},$$

and obtain an exact formula for it, where $p$ is an odd prime with $p \equiv 1 \pmod{h}$.

For any prime $p$ with $p \equiv 1 \pmod{6}$ and any non-principal character $\chi$ modulo $p$, whether there is an exact calculation formula for the $2k$-th power mean

$$\sum_{n=1}^{p-1} \left| \sum_{a=0}^{p-1} \chi\left(a^3 + n\right) \right|^{2k}, \ k \geq 3.$$

This is an open problem. We will study it further. Of course, if $\chi = \lambda$ is a three-order character modulo $p$, then from the methods of proving Theorem 2, we can prove that this conclusion is correct.

Theorem 3 shows a close relation between the quadratic residue in the form of $a^3 + 2$ modulo $p$ ($p \equiv 1 \pmod{6}$) and $d$. For example, if $(-1)^{\frac{p^2-1}{8}} \cdot d > 0$, then in a complete residue system modulo $p$, the number of quadratic residues in the form of $a^3 + 2$ is greater than the number of quadratic non-residues. If we take $p = 7, 13$ and $19$, then $(-1)^{\frac{49-1}{8}} \cdot 1 > 0$, $(-1)^{\frac{169-1}{8}} \cdot 1 < 0$ and $(-1)^{\frac{361-1}{8}} \cdot 1 < 0$, so we have

$$\sum_{a=0}^{6} \left( \frac{a^3 + 2}{p} \right) > 0, \ \sum_{a=0}^{12} \left( \frac{a^3 + 2}{p} \right) < 0, \ \sum_{a=0}^{18} \left( \frac{a^3 + 2}{p} \right) < 0.$$

This also describes the distribution properties of the quadratic residue in the form of $a^3 + 2$ modulo $p$ from a different perspective.

## 2. Several Lemmas

In this section, we provide several lemmas. Of course, for the proofs of these lemmas, some knowledge of analytic number theory is required. They can be found in many number theory books, such as [14–16]; here, we do not need to list these. Firstly, we have the following lemmas.

**Lemma 1.** *Let $p$ be a prime with $p \equiv 1 \pmod{6}$. Then, for integer $n$ with $(n, p) = 1$ and any non-principal character $\chi$ mod $p$, if $\chi^3 \neq \chi_0$, then we have the identity*

$$\left| \sum_{a=0}^{p-1} \chi\left(a^3 + n\right) \right| = \frac{1}{\sqrt{p}} \cdot \left| \lambda(n)\tau(\lambda)\tau(\overline{\chi}\overline{\lambda}) + \overline{\lambda}(n)\tau(\overline{\lambda})\tau(\overline{\chi}\lambda) \right|;$$

*If $\chi = \lambda$ is a third-order character modulo $p$, then we have*

$$\left| \sum_{a=0}^{p-1} \lambda\left(a^3 + n\right) \right| = \left| \frac{\lambda(n) \cdot \tau^3(\lambda)}{p} - \overline{\lambda}(n) \right|.$$

**Proof.** Let $\lambda$ be any third-order character modulo $p$. Then, for any integer $a$ with $(a, p) = 1$, note that the identity $1 + \lambda(a) + \overline{\lambda}(a) = 3$, if there is an integer $b$ with $(b, p) = 1$ such that $a \equiv b^3 \bmod p$; $1 + \lambda(a) + \overline{\lambda}(a) = 0$, otherwise. So, from these and the properties of Gauss sums we have

$$\sum_{a=0}^{p-1} \chi\left(a^3 + n\right) = \chi(n) + \sum_{a=1}^{p-1} \left(1 + \lambda(a) + \overline{\lambda}(a)\right)\chi(a + n)$$

$$= \sum_{a=0}^{p-1} \chi(a + n) + \sum_{a=1}^{p-1} \left(\lambda(a) + \overline{\lambda}(a)\right)\chi(a + n)$$

$$= \frac{1}{\tau(\overline{\chi})} \left( \sum_{b=1}^{p-1} \overline{\chi}(b) \sum_{a=1}^{p-1} \lambda(a) e\left(\frac{b(a + n)}{p}\right) + \sum_{b=1}^{p-1} \overline{\chi}(b) \sum_{a=1}^{p-1} \overline{\lambda}(a) e\left(\frac{b(a + n)}{p}\right) \right)$$

$$= \frac{\chi(n)}{\tau(\overline{\chi})} \left( \lambda(n)\tau(\lambda)\tau(\overline{\chi}\overline{\lambda}) + \overline{\lambda}(n)\tau(\overline{\lambda})\tau(\overline{\chi}\lambda) \right). \tag{2}$$

If $\chi = \lambda$, then from (2), $\tau(\overline{\chi}\lambda) = -1$, $\lambda^2 = \overline{\lambda}$ and $\tau(\lambda)\tau(\overline{\lambda}) = p$ we have

$$\left| \sum_{a=0}^{p-1} \lambda\left(a^3 + n\right) \right| = \left| \frac{\lambda(n) \cdot \tau^3(\lambda)}{p} - \overline{\lambda}(n) \right|. \tag{3}$$

Now, Lemma 1 follows from (2) and (3). $\quad\square$

**Lemma 2.** *Let $p$ be a prime with $p \equiv 1 \pmod 6$. Then, for any third-order character $\lambda$ mod $p$, we have the identity*

$$\tau(\chi_2) \cdot \tau(\overline{\lambda}) = \overline{\lambda}(2) \cdot \tau(\lambda\chi_2) \cdot \tau(\lambda),$$

*where $\tau(\chi) = \sum_{a=1}^{p-1} \chi(a)e\left(\dfrac{a}{p}\right)$ denotes the classical Gauss sums, $\left(\dfrac{*}{p}\right) = \chi_2$ denotes the Legendre's symbol modulo $p$, $e(y) = e^{2\pi i y}$ and $i^2 = -1$.*

**Proof.** Let $\psi = \lambda\chi_2$, then $\psi$ must be a sixth-order character modulo $p$. Note that $\psi^2 = \lambda^2 = \overline{\lambda}$, from the properties of the classical Gauss sums, we have

$$\sum_{a=0}^{p-1} \psi\left(a^2 - 1\right) = \sum_{a=0}^{p-1} \psi\left((a+1)^2 - 1\right) = \sum_{a=1}^{p-1} \psi\left(a^2 + 2a\right) = \sum_{a=1}^{p-1} \psi(a)\psi(a+2)$$

$$= \frac{1}{\tau(\overline{\psi})} \sum_{a=1}^{p-1} \psi(a) \sum_{b=1}^{p-1} \overline{\psi}(b)e\left(\frac{b(a+2)}{p}\right) = \frac{1}{\tau(\overline{\psi})} \sum_{b=1}^{p-1} \overline{\psi}(b) \sum_{a=1}^{p-1} \psi(a)e\left(\frac{b(a+2)}{p}\right)$$

$$= \frac{\tau(\psi)}{\tau(\overline{\psi})} \sum_{b=1}^{p-1} \overline{\psi}(b)\overline{\psi}(b)e\left(\frac{2b}{p}\right) = \frac{\tau(\psi)}{\tau(\overline{\psi})} \sum_{b=1}^{p-1} \lambda(b)e\left(\frac{2b}{p}\right) = \frac{\overline{\lambda}(2)\tau(\lambda)\tau(\psi)}{\tau(\overline{\psi})}. \tag{4}$$

On the other hand, note that for any integer $b$ with $(b, p) = 1$, we have the identity

$$\sum_{a=0}^{p-1} e\left(\frac{ba^2}{p}\right) = 1 + \sum_{a=1}^{p-1} (1 + \chi_2(a))e\left(\frac{ba}{p}\right) = \sum_{a=1}^{p-1} \chi_2(a)e\left(\frac{ba}{p}\right) = \chi_2(b) \cdot \tau(\chi_2),$$

so we also have the identity

$$\sum_{a=0}^{p-1} \psi\left(a^2 - 1\right) = \frac{1}{\tau(\overline{\psi})} \sum_{a=0}^{p-1} \sum_{b=1}^{p-1} \overline{\psi}(b)e\left(\frac{b(a^2 - 1)}{p}\right)$$

$$= \frac{1}{\tau(\overline{\psi})} \sum_{b=1}^{p-1} \overline{\psi}(b)e\left(\frac{-b}{p}\right) \sum_{a=0}^{p-1} e\left(\frac{ba^2}{p}\right) = \frac{\tau(\chi_2)}{\tau(\overline{\psi})} \sum_{b=1}^{p-1} \overline{\psi}(b)\chi_2(b)e\left(\frac{-b}{p}\right)$$

$$= \frac{\tau(\chi_2)}{\tau(\overline{\psi})} \sum_{b=1}^{p-1} \overline{\lambda}(b)e\left(\frac{-b}{p}\right) = \frac{\tau(\chi_2) \cdot \tau(\overline{\lambda})}{\tau(\overline{\psi})}. \tag{5}$$

Now, combining identities (4) and (5), we have

$$\tau(\chi_2) \cdot \tau(\overline{\lambda}) = \overline{\lambda}(2) \cdot \tau(\lambda\chi_2) \cdot \tau(\lambda).$$

This proves Lemma 2. $\quad\square$

**Lemma 3.** *Let $p$ be a odd prime with $p \equiv 1 \pmod 3$. Then, for any third-order character $\lambda$ mod $p$, we have*

$$\tau^3(\lambda) + \tau^3(\overline{\lambda}) = dp,$$

*where $d$ is uniquely determined by $4p = d^2 + 27b^2$ and $d \equiv 1 \pmod 3$.*

**Proof.** This result can be found in [6] or [10]. $\quad\square$

### 3. Proofs of the Theorems

In this section, we shall prove our main results. Firstly, we prove Theorem 1. If $\chi \neq \lambda$, then note that the identity

$$\sum_{n=1}^{p-1} \lambda(n) = \sum_{n=1}^{p-1} \overline{\lambda}(n) = 0,$$

from Lemma 1, we have

$$\left| \sum_{a=0}^{p-1} \chi\left(a^3 + n\right) \right|^2 = \frac{1}{p} \cdot \left( 2p^2 + \overline{\lambda}(n)\tau^2(\lambda)\tau(\overline{\chi}\overline{\lambda})\tau(\chi\overline{\lambda}) + \lambda(n)\tau^2(\overline{\lambda})\tau(\chi\lambda)\tau(\overline{\chi}\lambda) \right)$$

and

$$\sum_{n=1}^{p-1} \left| \sum_{a=0}^{p-1} \chi\left(a^3 + n\right) \right|^4$$

$$= \frac{1}{p^2} \cdot \sum_{n=1}^{p-1} \left( 2p^2 + \overline{\lambda}(n)\tau^2(\lambda)\tau(\overline{\chi}\overline{\lambda})\tau(\chi\overline{\lambda}) + \lambda(n)\tau^2(\overline{\lambda})\tau(\chi\lambda)\tau(\overline{\chi}\lambda) \right)^2$$

$$= 4p^2(p-1) + 2p^2(p-1) = 6p^2(p-1). \tag{6}$$

If $\chi = \lambda$, then from Lemma 1, we have

$$\sum_{n=1}^{p-1} \left| \sum_{a=0}^{p-1} \lambda\left(a^3 + n\right) \right|^4 = \sum_{n=1}^{p-1} \left( p + 1 - \frac{\overline{\lambda}(n) \cdot \tau^3(\lambda)}{p} - \frac{\lambda(n) \cdot \tau^3(\overline{\lambda})}{p} \right)^2$$

$$= (p+1)^2(p-1) + 2p(p-1) = (p-1)\left( p^2 + 4p + 1 \right). \tag{7}$$

Now, Theorem 1 follows from (6) and (7).

Now, we prove Theorem 2. From (3) and Lemma 3, we have

$$\sum_{n=1}^{p-1} \left| \sum_{a=0}^{p-1} \lambda\left(a^3 + n\right) \right|^6 = \sum_{n=1}^{p-1} \left( p + 1 - \frac{\overline{\lambda}(n) \cdot \tau^3(\lambda)}{p} - \frac{\lambda(n) \cdot \tau^3(\overline{\lambda})}{p} \right)^3$$

$$= (p+1)^3(p-1) - 3(p+1)^2 \sum_{n=1}^{p-1} \left( \frac{\overline{\lambda}(n) \cdot \tau^3(\lambda)}{p} + \frac{\lambda(n) \cdot \tau^3(\overline{\lambda})}{p} \right)$$

$$+ 3(p+1) \sum_{n=1}^{p-1} \left( \frac{\overline{\lambda}(n) \cdot \tau^3(\lambda)}{p} + \frac{\lambda(n) \cdot \tau^3(\overline{\lambda})}{p} \right)^2 - \sum_{n=1}^{p-1} \left( \frac{\overline{\lambda}(n) \cdot \tau^3(\lambda)}{p} + \frac{\lambda(n) \cdot \tau^3(\overline{\lambda})}{p} \right)^3$$

$$= (p+1)^3(p-1) + 6(p+1)p(p-1) - (p-1)\left( \frac{\tau^9(\lambda)}{p^3} + \frac{\tau^9(\overline{\lambda})}{p^3} \right)$$

$$= \left( p^2 - 1 \right)\left( p^2 + 8p + 1 \right) - \frac{p-1}{p^3} \left[ \left( \tau^3(\lambda) + \tau^3(\overline{\lambda}) \right)^3 - 3p^3\left( \tau^3(\lambda) + \tau^3(\overline{\lambda}) \right) \right]$$

$$= \left( p^2 - 1 \right)\left( p^2 + 8p + 1 \right) - (p-1)\left( d^3 - 3pd \right).$$

This proves Theorem 2.

To prove Theorem 3, we let $\lambda$ be a third-order character modulo $p$, then from the properties of Gauss sums (see Theorem 8.19 of [15]) and the identity

$$\sum_{a=0}^{p-1} \chi_2(a + m) = \sum_{b=0}^{p-1} \chi_2(b) = 0$$

we have

$$\sum_{a=0}^{p-1}\left(\frac{a^3+2}{p}\right) = \chi_2(2) + \sum_{a=1}^{p-1}\left(1+\lambda(a)+\overline{\lambda}(a)\right)\chi_2(a+2)$$

$$= \sum_{a=0}^{p-1}\chi_2(a+2) + \sum_{a=1}^{p-1}\left(\lambda(a)+\overline{\lambda}(a)\right)\chi_2(a+1)$$

$$= \frac{1}{\tau(\chi_2)}\sum_{a=1}^{p-1}\lambda(a)\sum_{b=1}^{p-1}\chi_2(b)e\left(\frac{b(a+2)}{p}\right) + \frac{1}{\tau(\chi_2)}\sum_{a=1}^{p-1}\overline{\lambda}(a)\sum_{b=1}^{p-1}\chi_2(b)e\left(\frac{b(a+2)}{p}\right)$$

$$= \frac{\chi_2(2)\cdot\lambda(2)\cdot\tau(\lambda)\cdot\tau(\chi_2\overline{\lambda}) + \chi_2(2)\cdot\overline{\lambda}(2)\cdot\tau(\overline{\lambda})\cdot\tau(\chi_2\lambda)}{\tau(\chi_2)}. \tag{8}$$

From Lemma 2, we have

$$\frac{\tau(\chi_2\lambda)}{\tau(\chi_2)} = \frac{\lambda(2)\cdot\tau(\overline{\lambda})}{\tau(\lambda)} \quad \text{and} \quad \frac{\tau(\chi_2\overline{\lambda})}{\tau(\chi_2)} = \frac{\overline{\lambda}(2)\cdot\tau(\lambda)}{\tau(\overline{\lambda})}. \tag{9}$$

Note that $\left(\frac{2}{p}\right) = (-1)^{\frac{p^2-1}{8}}$ and $\tau(\lambda)\tau(\overline{\lambda}) = p$, from (8), (9) and Lemma 3, we have

$$\sum_{a=0}^{p-1}\left(\frac{a^3+2}{p}\right) = \frac{\chi_2(2)\cdot\lambda(2)\cdot\tau(\lambda)\cdot\tau(\chi_2\overline{\lambda}) + \chi_2(2)\cdot\overline{\lambda}(2)\cdot\tau(\overline{\lambda})\cdot\tau(\chi_2\lambda)}{\tau(\chi_2)}$$

$$= \left(\frac{2}{p}\right)\cdot\frac{\tau^3(\lambda)+\tau^3(\overline{\lambda})}{p} = (-1)^{\frac{p^2-1}{8}}\cdot d.$$

This proves Theorem 3.

Now, we prove Corollary 1. If 2 is a cubic residue modulo $p$, then the congruence equation $x^3 + 2 \equiv 0 \pmod{p}$ has three solutions. So from Theorem 2, we know that $d$ is an even number. Since from Theorem 2, we have

$$\sum_{a=0}^{p-1}\left(\frac{a^3+2}{p}\right) = (-1)^{\frac{p^2-1}{8}}\cdot d, \tag{10}$$

and the left-hand side in (10) is an even number, so $d$ must be an even number.

If $d$ is an even number, then the left hand side in (10) must be an even number. So there is an integer $a$ such that $a^3 + 2 \equiv 0 \pmod{p}$. Note that $p \equiv 1 \pmod{3}$, so the congruence equations $x^3 \equiv -2 \pmod{p}$ and $x^3 \equiv 2 \pmod{p}$ must have three integer solutions. Thus, 2 must be a cubic residue modulo $p$.

This proves Corollary 1.

Now, we prove Corollary 2. From (1) we know that

$$4p = d^2 + 27b^2. \tag{11}$$

If 2 is a cubic residue modulo $p$, then $d$ is an even number. From (11), we know that $b$ must be an even number. Let $d = 2d_1$ and $b = 2b_1$, since $d \equiv 1 \pmod{3}$, so $d_1 \equiv -1 \pmod{3}$. Thus, from (11), we have the identity

$$p = d_1^2 + 27\cdot b_1^2,$$

where $d_1 \equiv -1 \pmod{3}$.

This completes the proofs of our main results.

## 4. Conclusions

If $p$ is a prime with $p \equiv 1 \pmod{6}$, then there must be two integers $d$ and $b$ such that the equation $4p = d^2 + 27b^2$, where $d$ is uniquely determined by $d \equiv 1 \pmod{3}$. The main

result of this paper is to give an exact calculation formula for the fourth power mean of one kind of character sum and an exact representation of $d$ by the Legendre's symbol modulo $p$. That is, we proved the identities

$$\frac{1}{p-1} \cdot \sum_{n=1}^{p-1} \left| \sum_{a=0}^{p-1} \chi \left( a^3 + n \right) \right|^4 = \left\{ \begin{array}{ll} 6p^2, & \text{if } \chi^3 \neq \chi_0; \\ p^2 + 4p + 1, & \text{if } \chi^3 = \chi_0 \end{array} \right.$$

and

$$\sum_{a=0}^{p-1} \left( \frac{a^3 + 2}{p} \right) = (-1)^{\frac{p^2-1}{8}} \cdot d.$$

From Theorem 3, we can deduce that 2 is a cubic residue modulo $p$ if and only if $d$ is an even number. This gives us a criterion for knowing that 2 is a cubic residue modulo $p$.

These results not only give the exact values of the character sums, they are also some new contribution to the research in related fields.

**Author Contributions:** Writing—original draft preparation, J.Z.; methodology, Y.M. All authors have read and agreed to the published version of the manuscript.

**Funding:** This research was funded by the N. S. F. (11771351) of P. R. China.

**Acknowledgments:** The authors would like to thank the editors and referees for their helpful suggestions and comments.

**Conflicts of Interest:** The authors declare that there are no conflict of interest regarding the publication of this paper.

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
