# Peer review of "The Mean Values of Character Sums and Their Applications"

_mathematics, doi:10.3390/math9040318_

Round 1

Reviewer 1 Report

In the paper the authors study  the calculating problems of some mean values of character sums of  special  polynomials.  Main results are given in Theorem 1 and Theorem 2. Using  elementary methods and  properties of the classical Gauss sums  that  results where proved. They also proved interesting lemmas and the proofs of these lemmas need some knowledge  of analytic number theory. The topic of the paper is interesting proofs seems clear and correct. The references are proper and actual.  I recommend publishing that paper in Mathematics.

Author Response

Dear  Editor,

We are very happy to receive your professional review comments, we have revised a little English spelling in our manuscript. Thank you very much for your work on our paper. 

Best regards,

Yuanyuan Meng, Jiafan Zhang

Reviewer 2 Report

The paper under review is well-written. Although, the Introduction should be extended. Precisely, the authors should highlight the importance and the novelty of their results. Also, some examples should be added.

Author Response

Dear  Editor,

We are very happy to receive your professional review comments, we have revised a little English spelling in our manuscript, and added some notes to  highlight the importance and the novelty of our results. At the same time, we also have added some examples in some notes.Thank you very much for your work on our paper. 

Best regards,

Yuanyuan Meng, Jiafan Zhang

Reviewer 3 Report

See attached file.

Author Response

This manuscript is a resubmission of an earlier submission. The following is a list of the peer review reports and author responses from that submission.

Round 1

Reviewer 1 Report

The authors should highlight the importance and novelty of their results. Also, several examples should be added.

Author Response

Dear Reviewer :

We are very grateful to you for your professional review comments, we have revised our paper according to your comments, the details of the revisions are as follows:

We added some notes before section 2(Several lemmas) to emphasize the importance of our results, gave some examples to show the significance of our results, and put forward some questions that we can continue to study in the future.

The above is all that we have modified, thank you again for your professional comments. If you have any queries, please don't hesitate to contact us at the address below.

Thank you and best regards.

Yours sincerely, 
Yuanyuan Meng

Reviewer 2 Report

Report of the paper The mean values of character and their applications

by Jiafan Zhang and Yuanyuan Meng the

In the paper the authors investigated the main values of character sums of special polynomials. Using  elementary methods and properties of the calssical Gauss sums they consider some calculating problems.

Results included in this paper are publishable, proofs seems correct and I recommend publishing that article in Mathematics.

Author Response

Dear Reviewer :

We are very grateful to you for your professional review comments and thank you very much for your work on our paper.

Thank you and best regards.

Yours sincerely, 
Yuanyuan Meng

Reviewer 3 Report

The paper under consideration looks correct, but I am not sure if it rises to the level of the journal. The results follow from standard techniques, and it looks more like a nice write up of an exercise than a research paper. In order to raise the level of the work, the author should have a really strong application of the material, or greatly expand the class of polynomials that can be studied. I thus do not recommend acceptance.

Author Response

Dear Reviewer :

We are very grateful to you for your professional review comments, we have revised our paper, the details of the revisions are as follows:

We added some notes before section 2(Several lemmas) to emphasize the importance of our results, gave some examples to show the significance of our results, and put forward some questions that we can continue to study in the future.

The above is all that we have modified, thank you again for your professional comments. If you have any queries, please don't hesitate to contact us at the address below.

Thank you and best regards.

Yours sincerely, 
Yuanyuan Meng
